# Amelioration of Systemic Amyloidosis by Blocking IL-17A and Not by IL-17F, and Arteriosclerosis by Blocking Both IL-17A and IL-17F in an Inflammatory Skin Mouse Model

**DOI:** 10.3390/ijms252111617

**Published:** 2024-10-29

**Authors:** Takehisa Nakanishi, Shohei Iida, Masako Ichishi, Makoto Kondo, Mai Nishimura, Ayaka Ichikawa, Yoshiaki Matsushima, Yoichiro Iwakura, Masatoshi Watanabe, Keiichi Yamanaka

**Affiliations:** 1Department of Dermatology, Mie University Graduate School of Medicine, 2-174 Edobashi, Tsu 514-8507, Japan; t-nakanishi@med.mie-u.ac.jp (T.N.); kmcasters@med.mie-u.ac.jp (S.I.); kondomak@med.mie-u.ac.jp (M.K.); ishika-m@clin.medic.mie-u.ac.jp (M.N.); ichika-a@med.mie-u.ac.jp (A.I.); matsushima-y@clin.medic.mie-u.ac.jp (Y.M.); 2Department of Oncologic Pathology, Mie University Graduate School of Medicine, 2-174 Edobashi, Tsu 514-8507, Japan; masako-i@doc.medic.mie-u.ac.jp (M.I.); mawata@doc.medic.mie-u.ac.jp (M.W.); 3Center for Animal Disease Models, Research Institute for Biomedical Sciences, Tokyo University of Science, Chiba 278-8510, Japan; iwakura@rs.tus.ac.jp

**Keywords:** inflammatory skin mouse model, psoriasis, atopic dermatitis, cytokine, IL-17A, IL-17F, IL-17AF, arteriosclerosis, organ amyloidosis

## Abstract

There are comorbidities and complications in atopic dermatitis and psoriasis that often occur after the appearance of skin inflammation. Statistically, data show that patients with psoriasis and atopic dermatitis have a shorter life expectancy than patients without psoriatic dermatitis, due to the occurrence of arteriosclerosis, myocardial infarction, and cerebral infarction. Many types of skin inflammation are treated with various antibody preparations, and marked improvement in patients’ quality of life can be achieved. The next theme is to understand the pathogenesis of arteriosclerosis, myocardial infarction, stroke, and other complications associated with dermatitis and to find treatments and drugs to reduce their occurrence. The skin, a crucial immune organ, generates large amounts of inflammatory cytokines in response to various stimuli, leading to systemic inflammation and potential damage to internal organs. The link between inflammatory skin conditions like psoriasis and atopic dermatitis with serious health complications such as vascular disorders and systemic amyloidosis has been increasingly recognized. In psoriasis, biological treatments targeting Interleukin (IL)-17A, a key cytokine, have shown promise in reducing cardiovascular risks. Recent developments include treatments that target both IL-17A and IL-17F in the psoriasis field, though each cytokine’s impact on internal organ damage is still under debate. Among visceral complications secondary to dermatitis, systemic amyloidosis and atherosclerosis have been reported to be controlled by suppressing IL-17 in the early stages of dermatitis. Still, it remains unclear whether suppressing IL-17 prevents organ damage in the late stages of persistent severe dermatitis. A study using a long-lasting dermatitis mouse model that overexpressed human caspase-1 in keratinocytes (Kcasp1Tg) investigated the effects of deleting IL-17A and IL-17F on visceral complications. Cross-mating Kcasp1Tg with IL-17A-, IL-17F-, and IL-17AF-deficient mice assessed the skin and visceral organs histologically, and RT-PCR analysis of aortic sclerosis markers was performed. Despite less improvement in dermatitis, deletion of IL-17A in Kcasp1Tg mice showed promising results in reducing multiple organ amyloidosis. On the other hand, the effect was observed in both IL-17A and IL-17F deleted mice for aortic sclerosis. The inhibition of IL-17A and IL-17F was suggested to reduce the risk of developing comorbidities in internal organs. IL-17A and IL-17F were found to act similarly or produce very different results, depending on the organ.

## 1. Introduction

It became statistically clear that intractable skin diseases such as psoriasis vulgaris and atopic dermatitis (AD) increase the risk for cerebro- and cardiovascular sclerosis, resulting in a higher incidence of cerebral and myocardial infarctions and, consequently, a shorter life span [1]. It is essential to pursue the association through studies in human and mouse models to determine why complications occur and to introduce treatment procedures. Data also prove that psoriasis and AD are associated with a higher frequency of other comorbidities and complications, such as osteoporosis, systemic amyloidosis, and liver fibrosis, which are also secondary to dermatitis. By suppressing cutaneous inflammation and skin-derived cytokines, these sequelae can be prevented. There is evidence that the incidence of cerebrovascular infarction was reduced by anti-cytokine therapies such as anti-TNF-α inhibitors [2]. Currently, there is an educational campaign to explain the public about the desirability of aggressive treatment for psoriasis and AD in order to prevent complications and comorbidities. However, the pathogenesis of complications and which cytokines play an essential role are still unknown, and more data need to be accumulated.

As an integral immune system component, the skin can respond to both exogenous and endogenous stimuli, potentially triggering systemic inflammation. Epidermal keratinocytes play a critical role, primarily by releasing stored cytokines, which activate the immune system and initiate a cytokine cascade. In the past, using an inflammatory mice model, IL-18 released from the skin by scratching, resulting in massive production of nonspecific IgE, and the crossbreeding of these mice with IL-18 knockout mice inhibited IgE production [3,4]. These results later led to the concept of innate lymphoid cell (ILC) type 2. After the onset of skin eruption, excessive cytokines produced enter the systemic circulation, impacting distant organs. Cardiovascular and cerebrovascular disorders [5,6], systemic amyloidosis [6,7,8], impaired sperm motility [9], osteoporosis [10], anxiety symptom [11], and even increased mortality risk at septic condition [12] may occur. In some cases, this process can lead to inflammation of fatty tissue, resulting in adipocytokine secretion, potentially affecting surrounding tissues such as the abdominal aorta [13]. Notably, systemic amyloidosis and vascular disease are serious complications associated with severe, intractable skin diseases, including psoriasis and AD [14,15,16]. This phenomenon is not limited to the single model mouse but also to another AD model that also develops the same internal amyloidosis and atherosclerosis [6]. The dermatitis triggers systemic inflammation [17].

Systemic amyloidosis is a disorder characterized by the accumulation of abnormal proteins in tissues and organs. It is often associated with chronic inflammatory diseases, such as rheumatoid arthritis [18,19,20] and inflammatory bowel disease [21,22,23,24,25]. These conditions can induce the liver-mediated overproduction of Serum Amyloid A (SAA) protein, leading to secondary AA protein amyloidosis [26,27]. Additionally, skin inflammation has been associated with amyloid deposition [6,7]. Elevated levels of SAA are positively correlated with the eosinophil count in the peripheral blood of patients with AD [28]. There is also a suspected link between severe dermatitis, gastrointestinal amyloidosis, and hypoalbuminemia, which may be due to protein leakage [7]. Furthermore, splenic amyloidosis can impair immune function and increase the risk of septic conditions attributable to hindered lymphocyte maturation [12].

Inflammatory cytokines, originating either from the skin or activated leukocytes, are believed to be significant contributors to the development of arteriosclerosis. In a clinical trial, endothelial function—measured by flow-mediated dilation (FMD)—exhibited improvement in patients with psoriasis following treatment with inhibitors targeting tumor necrosis factor-alpha (TNF-α) and Interleukin (IL)-17A [29,30]. Furthermore, IL-17 ablation led to improvements in arteriosclerosis in the dermatitis mice in the early phase, though there was considerable variability among individual subjects [31]. Consequently, it becomes imperative to gather detailed information regarding the duration of dermatitis and the point at which organ damage becomes irreversible.

IL-17 is a pivotal cytokine in the pathogenesis of psoriasis, and antibodies that target IL-17A have been effectively employed in treatment. Recently, therapeutic agents designed to target not only IL-17A but also IL-17F have been introduced, demonstrating promising outcomes in clinical settings [32,33]. The presence of IL-17A and IL-17F is essential for physiological processes; therefore, limiting their inhibition to the necessary extent is advisable. Nonetheless, in light of potential visceral complications, the necessity of suppressing both IL-17A and IL-17F remains uncertain. This ambiguity arises from a range of conflicting data in the literature [34,35,36].

To investigate the issue, we utilized a spontaneous inflammatory skin model, namely the keratin-14-driven caspase-1 transgenic (KCASP1Tg) mice [3,4,37]. KCASP1Tg mice began exhibiting initial dermatitis symptoms on their faces at approximately eight weeks of age, which then spread to the entire body without external stimuli (the details of the symptoms and histopathology are described in the Materials and Methods section). In this model, a reduction in the diameter of the abdominal aorta was observed, alongside increased mRNA levels indicative of aortic sclerosis at the initial phase [31]. Additionally, there was an overproduction of SAA1 and SAA2, with subsequent deposition of these proteins in the liver, spleen, and kidneys [6,8].

In the present study, to assess the effectiveness of targeting IL-17A and IL-17F in treating systemic amyloidosis and arteriosclerosis, KCASP1Tg mice were cross-mated with IL-17A-, IL-17F-, and IL-17AF-deficient mice, and observed until late stage of dermatitis.

## 2. Results

### 2.1. The Amyloid Deposition in the Liver Was Ameliorated by the Deletion of IL-17A and Not by IL-17F

Livers were collected and stained with hematoxylin and eosin (H&E), as well as direct fast scarlet (DFS) to examine amyloid deposition. Severe amyloid deposition was detected in the livers of KCASP1Tg mice but not in wild-type (WT) mice. Histological examination revealed that amyloid depositions radiated to the central veins from the Glisson’s capsule, including the interlobular bile ducts, interlobular arteries, and interlobular veins (Figure 1a). In KCASP1Tg mice, the livers were markedly enlarged compared to WT controls (Figure 1b). The liver was also enlarged in IL-17F-/KCASP1Tg mice. There was a significant increase in the DFS staining positive area in KCASP1Tg mice (Figure 1c,d). Amyloidosis in the liver was recovered in IL-17A-/KCASP1Tg mice and IL-17AF-/KCASP1Tg mice but not in IL-17F-/KCASP1Tg mice (Figure 1c,d). The serum liver transaminase aspartate aminotransferase (AST) and alanine aminotransferase (ALT) levels were increased in KCASP1Tg mice, which was ameliorated in IL-17A-/KCASP1Tg (Figure 1e).

### 2.2. The Amyloid Deposition in the Spleen Was Rescued by the Deletion of IL-17A and Restored Lymph Follicle Formation, Not by IL-17F Deletion

In KCASP1Tg mice, amyloid deposition was primarily observed in the marginal zone, the boundary between the white and red pulps (Figure 2a). The spleens were markedly enlarged in KCASP1Tg mice, and IL-17F-/KCASP1Tg mice also showed an enlarged spleen (Figure 2b). Severe amyloid deposition was also detected in the spleen of IL-17F-/KCASP1Tg mice. Amyloidosis in the spleen was ameliorated in IL-17A-/KCASP1Tg and IL-17AF-/KCASP1Tg mice. Immunostaining revealed decreased CD4, CD8, and CD20 cells in KCASP1Tg and IL-17F-/KCASP1Tg mice. CD138-positive cells were detected in KCASP1Tg mice and IL-17 deleted KCASP1Tg mice (Figure 2c). The positive DFS staining area showed a significant increase in KCASP1Tg mice, which decreased in IL-17A-/KCASP1Tg and IL-17AF-/KCASP1Tg mice (Figure 2d).

### 2.3. The Amyloid Deposition in the Kidney Was Lessened, and the Glomerular Constitution Was Restored by the Deletion of IL-17A but Not by IL-17F

The glomeruli and renal tubules were damaged in the kidneys of KCASP1Tg, and massive amyloid deposition was detected (Figure 3a). The kidneys in KCASP1Tg and IL-17F-/KCASP1Tg mice were markedly enlarged (Figure 3b). Amyloidosis was ameliorated in IL-17A-/KCASP1Tg and IL-17AF-/KCASP1Tg mice (Figure 3c,d). Urine protein and blood were also detected in KCASP1Tg mice, rescued in IL-17A-/KCASP1Tg mice but not in IL-17F-/KCASP1Tg mice (Figure 3e).

### 2.4. The Abdominal Arteriosclerosis Was Ameliorated by the Deletion of IL-17A and IL-17F

The histological analysis of the thoracic and abdominal aorta was performed. The arterial stricture was undetected in the thoracic aorta in all observed mice. Still, the stricture was detected in the abdominal aorta of KCASP1Tg mice compared to WT mice at six months (Figure 4a). No specific structural abnormalities were detected on the abdominal aorta. Measurement of the circumference of the lumen of the abdominal artery showed a predominant decrease in KCASP1Tg mice compared to WT littermates (Figure 4b), which were recovered by crossing with IL-17A-, IL-17F-, and IL-17AF- mice. Unfortunately, the increased and significant amelioration of mRNA expression level for Apol11b, Camp, Chil3, S100a8, and S100a9 were undetected in KCASP1Tg mice and other mice group measured (Figure 4c).

## 3. Discussion

There are comorbidities and complications in atopic dermatitis and psoriasis that often occur after the appearance of skin inflammation. Statistically, data show that patients with psoriasis have a 6-year shorter life expectancy than patients without psoriatic skin inflammation in terms of arteriosclerosis, myocardial infarction, and cerebral infarction [1]. We, as dermatologists, are trying to understand the pathogenesis of arteriosclerosis, myocardial infarction, stroke, and other complications and to find out how to treat them and which drugs to use to reduce their incidence. In addition to serving as a barrier against external pathogens, the skin is one of the largest immune organs, and it acts as an alarmin, releasing various inflammatory and pro-inflammatory cytokines in response to both external and intrinsic stimuli. Moreover, leukocytes infiltrate skin inflammation lesions and release a profusion of cytokines. The current study explored whether targeted suppression of a key cytokine, IL-17, could mitigate systemic amyloidosis and arteriosclerosis associated with skin inflammation. IL-17A and IL-17F are critical cytokines in the pathogenesis, especially in psoriasis vulgaris [38,39]. Usually, in most cases of psoriasis, inhibition of IL-17A alone results in marked improvement of the skin rash. In some cases, however, inhibition of IL-17A and IL-17F may also be necessary. The data show that inhibition of IL-17A improves psoriasis-associated complications, such as atherosclerosis. However, the specific reason for this improvement has yet to be determined. In addition, no data or findings have been published on the extent to which inhibition of IL-17A or IL-17F contributes to the suppression of arteriosclerosis. In AD, the predominant cytokines are IL-4, IL-5, and IL-13, and the pathogenesis differs from that of psoriasis. However, the skin inflammation in AD is a mixture of many inflammatory cytokines, including IL-17A and IL-17F. AD and psoriasis also share many comorbidities. KCASP1Tg mice, the mouse model used in this study, combine the pathogenesis of AD and psoriasis, and it is highly significant to examine the effect of IL-17 deficiency on the suppression of complications in this model. We observed organ-specific improvements through the inhibition of IL-17A and IL-17F.

We initially examined the correlation between skin inflammation and amyloid deposition in organs. Persistent, refractory skin inflammation may lead to organ amyloidosis [6,7]. A plethora of cytokines, released from areas of sustained inflammation, may enter the bloodstream, prompting the liver to synthesize amyloid A protein. Serum amyloid A (SAA) is a member of the apolipoprotein family, constitutively produced in various organs, and is also an acute-phase protein that is either induced or augmented by inflammatory stimuli, including proinflammatory cytokines such as IL-1, IL-6, and TNF-α, predominantly in the liver [40,41]. In the context of skin inflammation, inflammatory cytokines such as TNF-α, IL-17A, and IL-22 induce keratinocytes to produce proinflammatory cytokines, including IL-1β, IL-6, TNF-α, and IL-23, which subsequently influence the liver to increase SAA production [42]. Prior research indicates that the progression of visceral amyloidosis secondary to skin inflammation can be mitigated by inhibiting IL-17A during the early and mild phase of the disease [8], as well as through the application of JAK inhibitors [7]. However, the impacts of prolonged IL-17A and F inhibition on persistent inflammation have yet to be reported.

Our study revealed that mice with skin eruptions developed amyloidosis in the liver, spleen, and kidneys, accompanied by functional deterioration. Initially, amyloid deposition in the liver during the early stages of skin inflammation was mild, and there was only a slight elevation in liver enzymes [8]. However, prolonged skin inflammation in mice led to significant amyloid deposition and notable tissue disruption. Recovery from hepatic amyloidosis was observed in IL-17A-/KCASP1Tg and IL-17AF-/KCASP1Tg mice but not in IL-17F-/KCASP1Tg mice (Figure 1). Furthermore, serum liver transaminase levels (AST and ALT) were elevated in KCASP1Tg mice, a trend that was ameliorated in IL-17A-/KCASP1Tg and IL-17AF-/KCASP1Tg mice (Figure 1). Liver amyloidosis was poorly ameliorated in IL-17F deficiency. The liver transaminase levels were mildly increased in IL-17F-/KCASP1Tg mice compared to KCASP1Tg mice. This is consistent with clinical experience in humans, where increased transaminases tend to occur later in the terminal stages of amyloidosis. These findings suggest that IL-17A, but not IL-17F, plays a pivotal role in this context. Previous research indicates that pan-JAK inhibition also reduces amyloid deposition, which seemingly contradicts the observation that IL-17A does not engage the JAK pathway. However, JAK inhibitors are also effective in treating psoriasis, potentially due to their ability to interrupt the skin inflammation-mediated inflammatory cycle and reduce the overproduction of IL-17A.

Severe amyloid deposition was also observed in the spleens of KCASP1Tg mice, leading to substantial damage to the architecture of lymph follicles. Prolonged disruption can inhibit lymphocyte maturation in the spleen, resulting in immunosuppression. While no existing reports directly examine splenic architecture in individuals with inflammatory skin diseases, the lymphocyte depletion observed in patients with refractory autoinflammation and autoimmunity may be attributable to splenic amyloidosis. Notably, IL-17A has demonstrated promising results in psoriasis, where it not only suppresses skin inflammation but also ameliorates amyloidosis in the spleen. In the kidneys, amyloid deposition was accompanied by hypertrophy, likely due to a compensatory mechanism. The amyloid replaced and disrupted the glomeruli, essential for filtering waste products. This pathological change may contribute to renal dysfunction observed in autoinflammatory or autoimmune conditions.

Recent discussions have highlighted a strong association between persistent skin inflammation and atherosclerosis. The severity of skin inflammation is closely linked to an increased incidence of atherosclerosis, myocardial infarction, and cerebrovascular disease. The previous research suggests that atherosclerosis related to inflammation does not necessarily involve atheroma formation but rather involves narrowing, reduced elasticity, and increased fragility of the arteries. This can be attributed to two main factors. Firstly, inflammatory cytokines produced at the site of dermatitis enter the bloodstream and directly cause endothelial cell injury. Although these cytokines are slightly detectable in actual mouse blood samples, their presence is absent in normal mice, implying that prolonged exposure may result in vascular endothelial damage. Interestingly, no stenosis was observed in the thoracic aorta, whereas notable stenosis was found in the abdominal aorta. Low concentrations of inflammatory cytokines reach the abdominal adipose tissue via the bloodstream, triggering visceral fat inflammation and the subsequent production of adipocytokines, a process known as lipo-inflammation. In mice, significant inflammatory cell infiltration and fat inflammation were noted in abdominal adipose tissue. Moreover, the fatty tissue surrounding the abdominal aorta directly induces cytokines to act on the aorta, leading to its stiffening [13,17,43,44,45,46]. From previous research, the narrowing of the artery was detected in the basilar arteries of mice, supporting the observed correlation with increased cerebrovascular disease in humans [5]. These phenomena are encapsulated in the term “inflammatory skin march” [17].

Previous investigations utilizing GeneChip and RT-PCR analysis have identified several key factors implicated in arteriosclerosis in mice, including Apolipoprotein L 11b (Apol11b), Cathelicidin antimicrobial peptide (Camp), Chitinase 3-like 3 (Chil3), S100 calcium-binding protein A8 (S100a8), and S100 calcium-binding protein A9 (S100a9). Apol11b, thought to play a role in lipid and cholesterol transport and metabolism [47], is expressed under inflammatory conditions in myeloid and endothelial cells [48]. Apol11b is found in high-density lipoprotein complexes that are central to cholesterol transport. The cholesterol content of membranes is also important in cellular processes [49]. Camp, a polypeptide stored in the lysosomes of macrophages, polymorphonuclear leukocytes, epithelial cells, and human keratinocytes [50,51], is crucial for the mammalian innate immune defense against bacterial infections [52]; it reacts to pathogens by disrupting cell membranes, and its expression is upregulated following trauma, inflammation, or infection. Chil3 belongs to the chitinase-like proteins (CLPs), markers of immune activation and pathology with largely unexplored functions [53]. CLPs are known to induce both pro- and anti-inflammatory cytokines and chemokines, potentially modulating the inflammatory tumor microenvironment [54]. The S100A8/A9 complex, comprising calcium-binding proteins S100a8 and S100a9, plays a significant role in inflammatory responses by promoting leukocyte recruitment and cytokine secretion [55]. Improvements in arterial narrowing were observed in KCASP1Tg mice when crossed with IL-17A, F, and A/F knock-out mice. As listed above, the critical signal in arteriosclerosis was increased in KCASP1Tg at four months of age and decreased in IL-17 deficiency. However, the persistence of aortic inflammation markers for as long as 6 months of age was not alleviated by IL-17 inhibition. This may indicate that the phenotype of arteriosclerosis had already been completed at six months of age in the inflamed mice.

## 4. Materials and Methods

### 4.1. Animals

Six-month-old female transgenic mice in which keratinocytes specifically overexpress the human caspase-1 gene with the K14 promoter, designated as KCASP1Tg mice, were used in this study [37]. KCASP1Tg mice began exhibiting initial erosion around their eyes at approximately eight weeks of age, which then spread to the ear and neck, and multiple ulcers were formed on the face, trunk, and extremities without external stimuli under specific pathogen-free conditions. Re-epithelialization with atrophic skin occurred, but erosion and ulceration quickly relapsed. The hair of the face and eyelids disappeared, and the eyes were covered with a fibrous membrane. After 16 weeks, multiple skin ulcers formed, the ear and the eyelids were deformed, and the facial hair and the extremities were lost, only leaving the integument with multiple scars. At the light microscopic level, the epidermis of KCASP1Tg did not show particular histological changes by week 6. The thick epidermis around the ulcers of 10-week-old KCASP1Tg revealed psoriasis-like changes, including parakeratosis. The dermis of the ulcers was characterized by infiltration of many mononuclear cells. Keratinocytes in the lesions showed eosinophilic necrosis with nuclear condensation, a feature of apoptotic keratinocytes. KCASP1Tg epidermis revealed some apoptotic cell death with eosinophilic necrosis and marked leukocyte infiltration in the dermis. The epidermis surrounding the ulcers was significantly thicker than nontransgene mouse skin. C57BL/6 littermate mice (WT) were used as controls. KCASP1Tg was also crossed with IL-17A, IL-17F, and IL-17AF knockout mice (IL-17A-/KCASP1Tg, IL-17F-/KCASP1Tg, and IL-17AF-/KCASP1Tg mice, respectively) [56].

### 4.2. Experimental Design

The mice were housed in an environmentally conditioned room at 21 ± 2 °C, with a 12:12 h light cycle, 60% humidity, and food and water available ad libitum. The maximum caging density was five mice from the same litter and sex starting from weaning. As bedding, spruce wood shavings (Lignocel FS-14; J. Rettenmaier und Soehne GmbH, Rosenberg, Germany) were provided. Mice were fed a standardized mouse diet (1314, Altromin, Lage, Germany) and provided drinking water ad libitum. All materials, including IVCs, lids, feeders, bottles, bedding, and water were autoclaved before use. Monitoring of animal health was performed throughout the experiment. All mice were euthanized with CO_2_ and then sacrificed and analyzed at six months of age. To ensure compliance with the ARRIVE guidelines, the sample size was determined using statistical power analysis, ensuring that the minimum number of animals was used while maintaining adequate power to detect statistically significant differences. In this study, five groups were compared, with each group containing five mice, resulting in a total of 25 animals. This sample size was calculated based on previous studies and statistical calculations to ensure that a sufficient number of mice were included to detect significant differences between groups, while minimizing the total number of animals used. By employing this approach, we aimed to balance the ethical considerations of animal use with the need for statistical rigor in our findings. Animal weights were measured and monitored as in the preceding study, with similar results [6]. The mean weight of the animals and a standard deviation at six months old were as follows: WT, 30.99 g (standard deviation: SD 4); KCASP1Tg, 25.67 g (SD 1); IL-17A-/KCASP1Tg, 31.74 g (SD 1); IL-17F-/KCASP1Tg, 31.47 g (SD 1); and IL-17AF-/KCASP1Tg mouse, 31.74 g (SD 0), respectively. Each group kept 10 animals, of which 5 were randomly used in the experiment. Animal care was performed according to current ethical guidelines, and the Mie University Board Committee approved the experimental protocol for Animal Care and Use (#22-39-5-1).

### 4.3. Blood Sampling and Clinical Chemistry Parameters

All mice were euthanized with CO_2_. Blood was sampled from the tail vein or by cardiac puncture, placed in a 1.5 mL tube containing heparin, and centrifuged (6000 rpm for 5 min) to separate the plasma. The collected plasma was stored at −80 °C until examination. The value of liver functions was measured with commercially available systems. The urine protein and blood were also measured.

### 4.4. Tissue Sampling, Observation of the Abdominal Aorta, and Histological Analysis

All mice were subjected to euthanasia with CO_2_. Thoracic and abdominal aorta, liver, kidney, and spleen specimens were fixed in 10% buffered neutral formaldehyde and embedded in paraffin. Histological sections were 6-µm thick and stained with hematoxylin and eosin (H&E) (*n* = 5, each group), and direct fast scarlet staining (DFS) was performed on the liver, kidney, spleen, and aorta and then analyzed using ImageJ (https://imagej.net/software/imagej/). The percentage of DFS staining positive lesions was calculated. CD138, CD4, CD8, and CD20 staining was performed to identify the spleen’s specific type of infiltrating cells. For the thoracic and abdominal aorta, an Elastica van Gieson stain was also performed.

### 4.5. Real-Time Polymerase Chain Reaction (Real-Time PCR)

RT-PCR was performed to measure the changes in mRNA levels in the thoracic and abdominal aorta. Total RNA was extracted using Tri Reagent (Molecular Research Center, Cincinnati, OH, USA). The RNA concentration was measured using a NanoDrop Lite spectrophotometer (Thermo Fisher Scientific, Waltham, MA, USA), and one µg total RNA was converted to cDNA using a High-Capacity RNA-to-cDNA Kit (Applied Biosystems, Foster City, CA, USA). The TaqMan Universal PCR Master Mix II with UNG (Applied Biosystems, Waltham, MA, USA) was used to measure the mRNA expression of apolipoprotein L 11b (Apol11b, Mm03992571_s1), cathelicidin antimicrobial peptide (Camp, Mm00438285_m1), chitinase-like 3 (Chil3, Mm00657889_mH), S100 calcium-binding protein A8 (S100a8, Mm00496696_g1), and S100 calcium-binding protein A9 (S100a9, Mm00656925_m1). Gryceraldehyde-3-phosphate dehydrogenase (GAPDH, Mm99999915_g1) was used as an internal control. All probes were purchased from Applied Biosystems, and amplification was performed in a LightCycler 96 System (Roche Diagnostics, Indianapolis, IN, USA). The cycling parameters were as follows: 50 °C for 120 s, 95 °C for 600 s, followed by 40 cycles of amplification at 95 °C for 15 s, and 60 °C for 60 s.

### 4.6. Statistical Analysis

Statistical analyses were performed using the PRISM software version 10 (GraphPad, San Diego, CA, USA). All groups were analyzed using a Kruskal–Wallis test—n.s; no significant. Differences were considered statistically significant at *p* < 0.05. *; *p* < 0.05. **.

## 5. Conclusions

This study elucidated the detrimental effects of excessive IL-17 on organ function. While the presence of IL-17 is crucial for physiological processes within the body, its overproduction can lead to significant organ damage. Therefore, it is imperative to promptly and precisely mitigate excessive IL-17 production to prevent such adverse outcomes. In psoriasis treatment, in addition to agents that inhibit IL-17A, agents that inhibit both IL-17A and IL-17F have been used, and both have shown marked efficacy in skin conditions. Poorly controlled cutaneous manifestations of psoriasis can lead to a variety of complications, often fatal. Until now, there have been no data on how cytokine-targeted antibody therapies control these complications. In the present study, inhibition of IL-17A was significantly effective against systemic amyloidosis, whereas inhibition of IL-17F was not. Both IL-17A and IL-17F inhibition showed preventive effects against arteriosclerosis. Further studies are needed to increase the number of samples and to examine the effects of IL-17A and IL-17F in other mouse models. In addition, it is necessary to accumulate data in humans before and after treatment with antibody therapy.

## Figures and Tables

**Figure 1 ijms-25-11617-f001:**
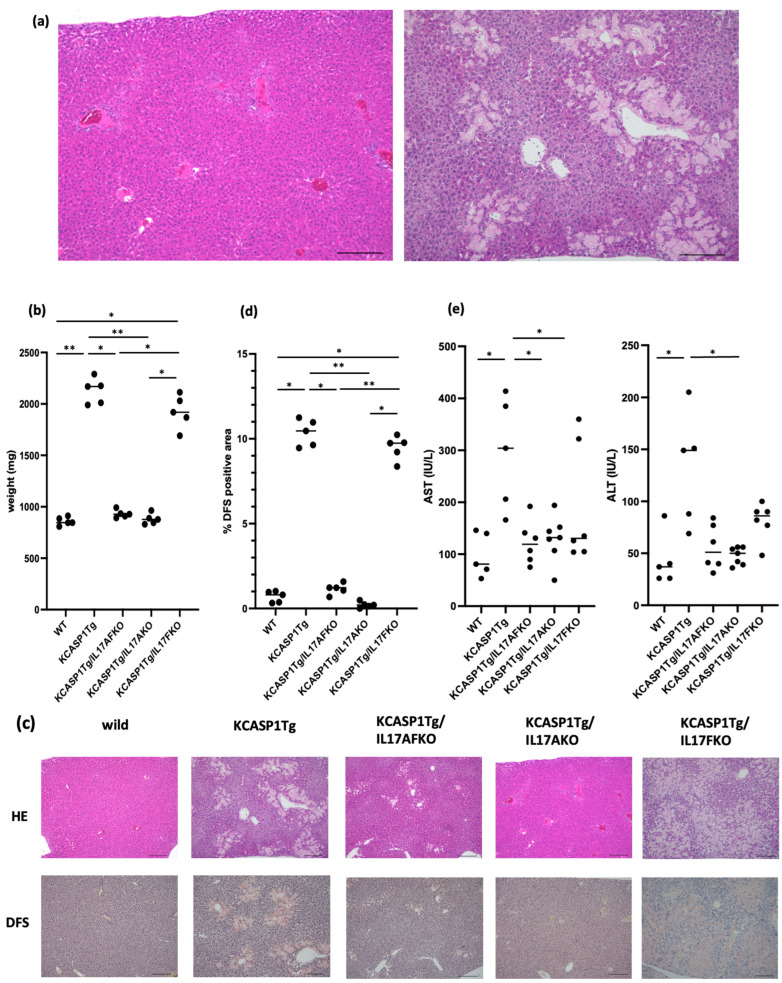
Histological analysis of the liver. The liver was collected from 6-month-old mice (n = 5, each) and stained with hematoxylin and eosin (H&E) and direct fast scarlet (DFS). (**a**) Severe amyloid deposition was detected in the livers of KCASP1Tg mice, but not in WT mice. In particular, amyloid was massively deposited radiating from the Glisson’s capsule, including the interlobular bile ducts, interlobular arteries, and interlobular veins, to the central veins of the liver (bar 200 μm). (**b**) In KCASP1Tg mice, the livers were markedly enlarged compared to WT controls. The weight was also high in IL-17F-deleted KCASP1Tg mice. (**c**,**d**) IL-17A deletion (IL-17A-/KCASP1Tg mice) rescued the amyloidosis dramatically, but not in IL-17F-deleted KCASP1Tg mice. IL-17AF-/KCASP1Tg mice showed decreased amyloid deposition (×100 magnification). The positive DFS staining area was significantly increased in KCASP1Tg and IL-17F-/KCASP1Tg mice. (**e**) The concentration of serum liver transaminase was elevated in KCASP1Tg mice with significance and was ameliorated by IL-17A deletion (AST, aspartate aminotransferase; ALT, alanine aminotransferase). The data are expressed as mean ± SD. All groups were analyzed using a Kruskal–Wallis test. * *p* < 0.05, ** *p* < 0.01.

**Figure 2 ijms-25-11617-f002:**
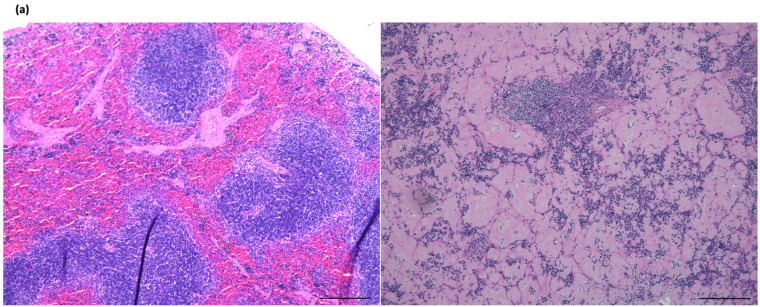
Histological analysis of spleen. (**a**) Severe amyloid deposition was detected in KCASP1Tg mice and not in 6-month-old WT mice. The marginal zone is the boundary between the white and red pulp, and dense deposition was detected (bar 200 μm). (**b**) The spleen was markedly enlarged in KCASP1Tg mice compared to WT controls. The weight was also high in IL-17F-/KCASP1Tg mice. (**c**) Amyloid deposition was rescued in IL-17A-/KCASP1Tg. Immunostaining for CD4, CD8, CD20, and CD138 was performed, and CD4-, CD8-, and CD20-positive cells were decreased in KCASP1Tg and IL-17F-/KCASP1Tg mice. On the contrary, CD138-positive cells were more stained in KCASP1Tg mice (×100 magnification). (**d**) The positive DFS staining area quantified by a Kruskal–Wallis test showed a significant increase in DFS positive area in KCASP1Tg mice, which decreased in IL-17A-/KCASP1Tg and IL-17AF-/KCASP1Tg mice. * *p* < 0.05, ** *p* < 0.01 and *** *p* < 0.001.

**Figure 3 ijms-25-11617-f003:**
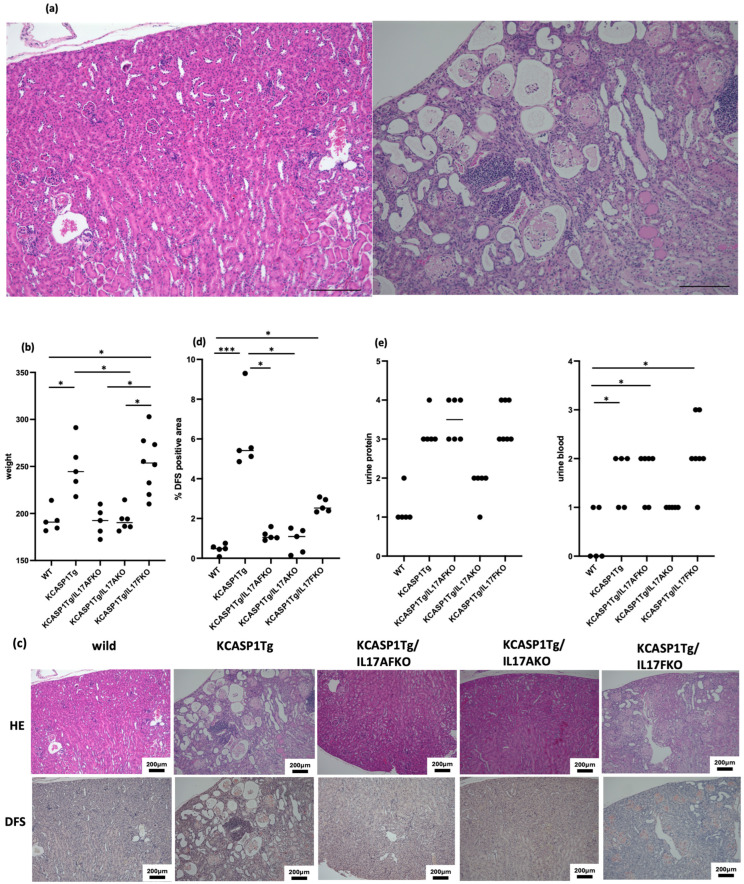
Histological analysis of spleen. (**a**) Amyloidosis in the kidney was severe, and amyloid replaced glomeruli in KCASP1Tg mice (bar 200 μm). (**b**) In KCASP1Tg mice, the kidneys were markedly enlarged compared to WT mice. (**c**,**d**) Amyloid deposition was ameliorated in IL-17A-/KCASP1Tg and IL-17AF-/KCASP1Tg mice. (**e**) Urine protein and blood were detected in KCASP1Tg mice, which were also rescued in IL-17A-/KCASP1Tg mice. * *p* < 0.05, *** *p* < 0.001.

**Figure 4 ijms-25-11617-f004:**
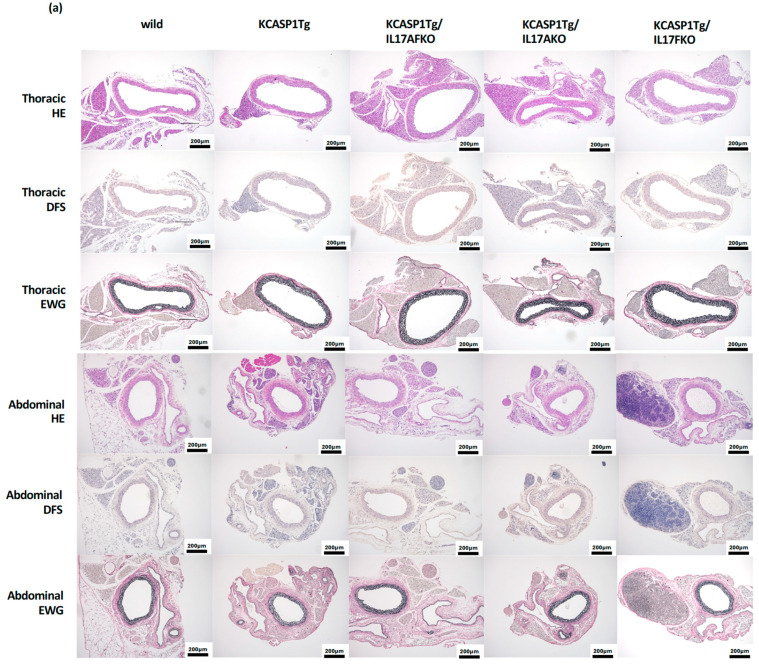
The abdominal arteriosclerosis was ameliorated by the deletion of IL-17A and IL-17F (**a**) The arterial stricture was undetected in the thoracic aorta in all observed mice. The narrowing of the artery was observed in KCASP1Tg mice at the abdominal aorta. There were no pathological abnormalities at the abdominal aorta in KCASP1Tg mice. (**b**) The artery’s lumen circumference measurement showed a predominant decrease in the abdominal aorta of KCASP1Tg compared to WT mice. To investigate the effect of IL-17A, F, and AF in arteriosclerosis, the circumference of the lumen of the artery was measured in IL-17A-/KCASP1Tg, IL-17F-/KCASP1Tg, and IL-17AF-/KCASP1Tg mice. The circumference of the lumen of the artery was recovered at the abdominal portion in KCASP1Tg mice when crossing with IL-17A-, IL-17F-, and IL-17AF- mice. (**c**) The increase and specific amelioration in mRNA expression level for the enhanced gene, Apol11b, Camp, Chil3, S100a8, and S100a9, were undetected in mice groups measured at six months of old (Upper panel; thoracic aorta and Lower panel; abdominal aorta). * *p* < 0.05.

## Data Availability

Data are contained within the article.

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
