# Peer review of "Amelioration of Systemic Amyloidosis by Blocking IL-17A and Not by IL-17F, and Arteriosclerosis by Blocking Both IL-17A and IL-17F in an Inflammatory Skin Mouse Model"

_ijms, 2024, doi:10.3390/ijms252111617_

Round 1
Reviewer 1 Report
Comments and Suggestions for Authors
It is an excellent study that analyzed the effects of IL-17A and IL-17F on systemic amyloidosis and arteriosclerosis in inflammatory skin mouse model.
Hereby please find my comments regarding the paper:
1. The abstract, the introduction and the results are well presented.
2. Explain the abbreviations when they appear for the first time in the text.
3. In the discussion`s section, please make a difference between the profile of inflammatory cytokine in psoriasis versus dermatitis. The IL17A and F profile it is found in psoriasis lesions, therefore please change the term dermatitis, maybe with skin inflammation. Also, the term “inflammatory skin march” is better associated with atopic dermatitis. In psoriasis it is better to use “psoriatic march”.
4. Material and methods: in 4.2 all the mice were euthanatized with CO2, but in 4.3 you say that you used CO2 or pentobarbital. What is the correct version?
5. The conclusions need to be more detailed.
6. The references are appropriate.
Author Response
It is an excellent study that analyzed the effects of IL-17A and IL-17F on systemic amyloidosis and arteriosclerosis in inflammatory skin mouse model.
Hereby please find my comments regarding the paper:
Comment 1. The abstract, the introduction and the results are well presented.
Response: Thank you very much for taking the time to review this report.
Comment 2. Explain the abbreviations when they appear for the first time in the text.
Response: Thank you for the suggestion. We have modified it.
Comment 3. In the discussion`s section, please make a difference between the profile of inflammatory cytokine in psoriasis versus dermatitis. The IL17A and F profile it is found in psoriasis lesions, therefore please change the term dermatitis, maybe with skin inflammation. Also, the term “inflammatory skin march” is better associated with atopic dermatitis. In psoriasis it is better to use “psoriatic march”.
Response: Thank you for your remarks. I described the differences in inflammatory cytokine profiles in psoriasis and atopic dermatitis in the discussion section. Usually, the IL17A and F profiles are more common in psoriasis lesions. However, in the lesional skin of atopic dermatitis, Type 2 cytokines such as IL-4, IL-5, and IL-13 are, of course, predominant, but inflammatory cytokines, including IL-17A and IL-17F, are very abundant and mixed. Atopic dermatitis and psoriasis also share many of the complications and comorbidities that occur when poorly controlled. Fortunately, the mice used in this study are a model that shares the skin manifestations and immunological profile of atopic dermatitis and psoriasis. Therefore, we looked at the inhibitory effect of IL-17 in these two diseases. The term psoriatic march is used for psoriasis, while the term inflammatory skin march refers to the vascular lesions commonly occurring in inflammatory skin diseases including both psoriasis and atopic dermatitis. Thank you for pointing this out.
Comment 4. Material and methods: in 4.2 all the mice were euthanatized with CO2, but in 4.3 you say that you used CO2 or pentobarbital. What is the correct version?
Response: CO2 is correct. We have modified the Material and Methods section.
Comment 5. The conclusions need to be more detailed.
Response: We have modified it. We appreciate your comment.
Comment 6. The references are appropriate.
Response: Thank you.
- We have supplemented the references. We appreciate your comments.

Reviewer 2 Report
Comments and Suggestions for Authors
This manuscript reports studies in which the expression of human caspase-1 in keratinocytes was used to induce dermatitis in a mouse model of psoriasis. By crossing with other mice, IL-17A, IL-17F, and both IL-17A and F were deleted, to investigate the roles of these cytokines in visceral complications. The authors show that the blockade of IL-17A reduces systemic amyloidosis whereas targeting IL-17F has no beneficial effect. Futhermore, the blockade of both IL-17A and IL-17F reduced arterioslerosis according to the interpretation of the authors.
The aims and the results of the study are largely interesting. However, the link between dermatitis and the changes in other organs is not clear, and the presentation of the results should be improved. In its current form, the manuscript is not convincing.
Specific comments:
The authors do not report the skin phenotypes of the mice. This is a major weakness of this study. More information about the skin is essential.
The title and the abstract of the manuscript do not fit together well. The abstract does not summarize the entire study. This needs to be improved.
The histological images are presented in a minimalistic way. It is strongly recommended to label some of the details in the images to focus the attention on the key parts.
Line 157: I am not sure what the authors want to say with the word “facilitated”.
The presentation of the data in section 2.4 is unclear. It is difficult to understand what is disease-relevant and what is significant here. Variation seems to be large.
Please explain why mice at an age six months were investigated.
Line 258: Mouse age was 4 months. Correct?
The importance of IL-17 for the skin should be explained better. More information on the role of IL-17 in cutaneous immune homeostasis and references to up-to-date reviews should be included in the introduction section.
Author Response
This manuscript reports studies in which the expression of human caspase-1 in keratinocytes was used to induce dermatitis in a mouse model of psoriasis. By crossing with other mice, IL-17A, IL-17F, and both IL-17A and F were deleted, to investigate the roles of these cytokines in visceral complications. The authors show that the blockade of IL-17A reduces systemic amyloidosis whereas targeting IL-17F has no beneficial effect. Futhermore, the blockade of both IL-17A and IL-17F reduced arterioslerosis according to the interpretation of the authors.
The aims and the results of the study are largely interesting. However, the link between dermatitis and the changes in other organs is not clear, and the presentation of the results should be improved. In its current form, the manuscript is not convincing.
Response: Thank you for your time and peer review. In the field of dermatology, there are comorbidities and complications in atopic dermatitis and psoriasis that often occur after the appearance of skin inflammation. Statistically, data show that patients with psoriasis and atopic dermatitis have a 6-year shorter life expectancy than patients without psoriatic dermatitis in terms of arteriosclerosis, myocardial infarction, and cerebral infarction. We, as dermatologists, are trying to understand the pathogenesis of atherosclerosis, myocardial infarction, stroke, and other complications and to find out how to treat them and which drugs to use to reduce their incidence. Our institution is the world leader in this field, with more publications; therefore, the number of citations is higher. As a guest editor, the theme and title of this project are related to this. Previous papers have shown that inhibition of IL-17A suppresses arteriosclerosis in mice with dermatitis in the early stage of the disease, but the current manuscript is a follow-up study of the so-called late onset of the disease. In addition, we now have conclusive data on which drugs should be used to prevent systemic amyloidosis. Although inhibitors of both IL-17A and IL-17F exist as drugs for the treatment of humans, it was found that inhibition of both IL-17A and IL-17F is not necessary for the treatment of amyloidosis, although inhibition of IL-17A is essential for the treatment of amyloidosis. This is a critical medical observation. We report this information because it is important for researchers, clinicians, and pharmaceutical manufacturers treating psoriasis and atopic dermatitis. However, we have changed the manuscript to make it easier for readers new to this area to understand. We appreciate your comments.
Specific comments:
Comment 1: The authors do not report the skin phenotypes of the mice. This is a major weakness of this study. More information about the skin is essential.
Response: KCASP1Tg mice began exhibiting initial erosion around their eyes at approximately eight weeks of age, which then spread tothe ear and neck, and multiple ulcers were formed on the face, trunk, and extremities without external stimuli under specific pathogen-free conditions. Re-epithelialization with atrophic skin occurred, but erosion and ulceration quickly relapsed. The hair of the face and eyelids disappeared, and the eyes were covered with a fibrous membrane. After 16 weeks, multiple skin ulcers formed, the ear and the eyelids were deformed, and the facial hair and the extremities were lost, only leaving the integument with multiple scars.
At the light microscopic level, the epidermis of KCASP1Tg did not show particular histological changes by week 6. The thick epidermis around the ulcers of 10-week-old KCASP1Tg revealed psoriasis-like changes, including parakeratosis. The dermis of the ulcers was characterized by infiltration of many mononuclear cells. Keratinocytes in the lesions showed eosinophilic necrosis with nuclear condensation, a feature of apoptotic keratinocytes. KCASP1Tg epidermis revealed some apoptotic cell death with eosinophilic necrosis and marked leukocyte infiltration in the dermis. The epidermis surrounding the ulcers was significantly thicker than non-transgene mouse skin. This information has been supplemented in the revised version.
Comment 2: The title and the abstract of the manuscript do not fit together well. The abstract does not summarize the entire study. This needs to be improved.
Response: We have modified the abstract as the reviewer pointed.
Comment 3: The histological images are presented in a minimalistic way. It is strongly recommended to label some of the details in the images to focus the attention on the key parts.
Response: We have modified and supplemented the figures.
Comment 4: Line 157: I am not sure what the authors want to say with the word “facilitated”.
Response: It was an error in word usage. It has been corrected.
Comment 5: The presentation of the data in section 2.4 is unclear. It is difficult to understand what is disease-relevant and what is significant here. Variation seems to be large.
Response: In this regard, we completely agree with the reviewer. We were especially hoping that the RT-PCR data would be clean and also clearly and significantly different. However, we assume this is due to the mix of mice that have already completed the inflammatory phase and are no longer showing mRNA activity due to the completion of dermatitis-derived arteriosclerosis at 6 months. This comment was supplemented in the discussion.
Comment 6: Please explain why mice at an age six months were investigated.
Response: Dermatitis in mice develops at the age of 2 months and reaches its peak at the age of 4 months. We have previously reported peak dermatitis, immune status, and associated organ lesions. We speculate that arteriosclerosis is underway due to the effects of the dermatitis. This time, we look at organ changes at 6 months, when the skin rash is complete and the lesions are probably fixed.
Comment 7: Line 258: Mouse age was 4 months. Correct?
Response: Yes, the reviewer is correct. At 4 months of age, we can assume that arteriosclerosis is in progress due to the effects of dermatitis, and thus the mediators involved in arteriosclerosis were elevated when RT-PCR was performed. However, at 6 months of age, these mediators were not elevated. At 6 months, arteriosclerosis has probably already been completed, and further progression is no longer possible. We have revised the content for clarity.
Comment 8: The importance of IL-17 for the skin should be explained better. More information on the role of IL-17 in cutaneous immune homeostasis and references to up-to-date reviews should be included in the introduction section.
Response: Thank you for pointing this out. IL-17A and IL-17F are critical cytokines in the pathogenesis, especially in psoriasis vulgaris. Usually, in most cases of psoriasis, inhibition of IL-17A alone results in marked improvement of the skin rash. In some cases, however, inhibition of IL-17F may also be necessary. In some cases, data show that inhibition of IL-17A improves sclerosis, especially in psoriasis-associated complications, such as atherosclerosis. However, the specific reason for this improvement has yet to be determined. In addition, no data or findings have been published on the extent to which inhibition of IL-17F contributes to the suppression of arteriosclerosis. This paper will probably be the first of its kind in the world. We have supplemented the references. We appreciate your comments.

Reviewer 3 Report
Comments and Suggestions for Authors
The present study comparatively examines the impact of the overexpression of human caspase-1 in keratinocytes (Kcasp1Tg) on systemic amyloidosis and atherosclerosis. Although the concept underlying the study appears promising, several aspects of the work seem to raise questions.
First of all, the title, abstract, and introduction make a connection between the present study and the skin. But, according to my understanding, this study is not related to the skin, except for the fact that the animal model used has previously been utilized by the authors as a model for simulating skin inflammation. However, neither the experiments, the design, nor the aim of the study pertain to the skin. Furthermore, the results do not draw any conclusions regarding the health or biology of the skin in these transgenic experimental animals. Therefore, the connection of this study to the skin, in my opinion, constitutes a logical leap and should be reconsidered, meaning that the entire study needs to be rewritten.
Secondly, in the materials and methods section, there are some significant omissions that prevent the reader from understanding the importance of the study. Regarding the experimental animals used, there is no mention of their number, the groups that were used, the origin of the experimental animals, as well as most of the details required by the ARRIVE (https://arriveguidelines.org/sites/arrive/files/documents/Author%20Checklist%20-%20Full.pdf.) guidelines, which, according to the journal, are mandatory for the papers it publishes. A complete record of the procedures concerning the use of experimental animals should be provided by the authors. Clarification should be provided also about the animal’s scarification methods since there are reported two different types of euthanasia methods (CO2 or Phenobarbital).
About the statistical methods concerns also may be raised. If I understand correctly (as it is not clearly described), this appears to be a study that used 5 experimental animals in each group. If this is true, then we are talking about a relatively low number of animals. In this case, and if a parametric method is to be used, the results should first be checked for normality of distribution (Shapiro-Wilk test), followed by homogeneity of variance (Levene's test). If everything is in order, only then can ANOVA be used. Otherwise, a non-parametric method should have been used.
Finally, an additional point that should be reconsidered, in my opinion, is the issue of self-citations. It seems that almost half of the references used in this manuscript pertain to authors who are also involved in the present writing. Although self-citation in this case could be more extensive than usual, given that this particular group appears to have done significant work with this animal model, the number of self-citations should nevertheless be reassessed.
Comments on the Quality of English LanguageModerate changes in English language may be needed
Author Response
The present study comparatively examines the impact of the overexpression of human caspase-1 in keratinocytes (Kcasp1Tg) on systemic amyloidosis and atherosclerosis. Although the concept underlying the study appears promising, several aspects of the work seem to raise questions.
Comment 1: First of all, the title, abstract, and introduction make a connection between the present study and the skin. But, according to my understanding, this study is not related to the skin, except for the fact that the animal model used has previously been utilized by the authors as a model for simulating skin inflammation. However, neither the experiments, the design, nor the aim of the study pertain to the skin. Furthermore, the results do not draw any conclusions regarding the health or biology of the skin in these transgenic experimental animals. Therefore, the connection of this study to the skin, in my opinion, constitutes a logical leap and should be reconsidered, meaning that the entire study needs to be rewritten.
Response: Thank you for your time and peer review. In the field of dermatology, there are comorbidities and complications in atopic dermatitis and psoriasis that often occur after the appearance of skin inflammation. Statistically, data show that patients with psoriasis and atopic dermatitis have a 6-year shorter life expectancy than patients without psoriatic dermatitis in terms of arteriosclerosis, myocardial infarction, and cerebral infarction. We, as dermatologists, are trying to understand the pathogenesis of atherosclerosis, myocardial infarction, stroke, and other complications and to find out how to treat them and which drugs to use to reduce their incidence. Our institution is the world leader in this field, with more publications; therefore, the number of citations is higher. As a guest editor, the theme and title of this project are related to this. Previous papers have shown that inhibition of IL-17A suppresses arteriosclerosis in mice with dermatitis in the early stage of the disease, but the current manuscript is a follow-up study of the so-called late onset of the disease. In addition, we now have conclusive data on which drugs should be used to prevent systemic amyloidosis. Although inhibitors of both IL-17A and IL-17F exist as drugs for the treatment of humans, it was found that inhibition of both IL-17A and IL-17F is not necessary for the treatment of amyloidosis, although inhibition of IL-17A is essential for the treatment of amyloidosis. This is a critical medical observation. We report this information because it is important for researchers, clinicians, and pharmaceutical manufacturers treating psoriasis and atopic dermatitis. However, we have changed the abstract, introduction, and discussion to make it easier for readers new to this area to understand. We appreciate your comments.
Comment 2: Secondly, in the materials and methods section, there are some significant omissions that prevent the reader from understanding the importance of the study. Regarding the experimental animals used, there is no mention of their number, the groups that were used, the origin of the experimental animals, as well as most of the details required by the ARRIVE (https://arriveguidelines.org/sites/arrive/files/documents/Author%20Checklist%20-%20Full.pdf.) guidelines, which, according to the journal, are mandatory for the papers it publishes. A complete record of the procedures concerning the use of experimental animals should be provided by the authors. Clarification should be provided also about the animal’s scarification methods since there are reported two different types of euthanasia methods (CO2 or Phenobarbital).
Response: We appreciate your comments. The details of animals used in this experiment has been described in the materials and methods section, 4.1.
Comment 3: About the statistical methods concerns also may be raised. If I understand correctly (as it is not clearly described), this appears to be a study that used 5 experimental animals in each group. If this is true, then we are talking about a relatively low number of animals. In this case, and if a parametric method is to be used, the results should first be checked for normality of distribution (Shapiro-Wilk test), followed by homogeneity of variance (Levene's test). If everything is in order, only then can ANOVA be used. Otherwise, a non-parametric method should have been used.
Response: Thank you for your comments. We have changed to non-parametric analysis, as the reviewer suggested.
Comment 4: Finally, an additional point that should be reconsidered, in my opinion, is the issue of self-citations. It seems that almost half of the references used in this manuscript pertain to authors who are also involved in the present writing. Although self-citation in this case could be more extensive than usual, given that this particular group appears to have done significant work with this animal model, the number of self-citations should nevertheless be reassessed.
Response: Thank you for pointing this out. This special issue aims to clarify the pathogenesis of complications and comorbidities derived from inflammatory skin diseases and to educate the public further. For us dermatologists, we are trying to tackle the mystery of why arteriosclerosis, cerebral infarction, myocardial infarction, and systemic amyloidosis occur more frequently in patients with psoriasis vulgaris and atopic dermatitis and which treatment methods are the most efficient and safe in the world. Our laboratory is leading the way. We are fortunate because we have developed several dermatitis models for atopic dermatitis and psoriasis vulgaris. Therefore, the number of self-citations has increased, and we have tried to reduce the percentage in consultation with the editorial office. We appreciate the reviewer’s comment.

Round 2
Reviewer 1 Report
Comments and Suggestions for Authors
Dear authors,
You took into account all the suggestions given.
Author Response
Thank you very much for taking the time to review this report.Reviewer 2 Report
Comments and Suggestions for Authors
Thank you for adding more information.
Author Response
Thank you very much for taking the time to review this report.Reviewer 3 Report
Comments and Suggestions for Authors
Dear authors, it is understandable that conditions like arteriosclerosis and heart attacks are more common in patients with skin diseases like psoriasis and atopic dermatitis. But, based on the data in this specific study, I don’t think there’s enough evidence to show a generalized link between skin inflammation and these systemic conditions. The KCASP1Tg mouse model does indeed show both skin and systemic effects, but the design of this experiment doesn’t seem to prove that skin inflammation is a cause of diseases like arteriosclerosis.
The present study mainly looks at how Caspase 1 acts by blocking IL-17A and in this way helps with systemic amyloidosis and how by blocking both IL-17A and IL-17F affects arteriosclerosis. These results seem more focused on inflammation in liver, spleen, kidney and aortic tissues, rather than proving that skin inflammation leads to systemic conditions. Without clear evidence of this connection, the link to skin inflammation feels weak.
I suggest reconsidering how much emphasis is placed on the skin in the title, abstract, and introduction. The real focus of your study seems not to be on the skin but on the liver, spleen, kidneys and aorta. Shifting the focus to better match your actual findings would make the study stronger and more accurate.
While i thank you for your response, i still think the manuscript should be revised to better reflect the data you’ve presented.
Also, about the ARRIVE guidelines the authors must add some more things in their materials and methods part. Especially
1. Sample size justification (Why 5 animals and not less or more?)
2. Weight of the animals (Providing also SD)
3. Randomization methods (If not state that here was no randomization)
4. Monitoring of animal’s health (for example Weight or temperature measurements throughout the experiment)
Author Response
Comment: Dear authors, it is understandable that conditions like arteriosclerosis and heart attacks are more common in patients with skin diseases like psoriasis and atopic dermatitis. But, based on the data in this specific study, I don’t think there’s enough evidence to show a generalized link between skin inflammation and these systemic conditions. The KCASP1Tg mouse model does indeed show both skin and systemic effects, but the design of this experiment doesn’t seem to prove that skin inflammation is a cause of diseases like arteriosclerosis.
The present study mainly looks at how Caspase 1 acts by blocking IL-17A and in this way helps with systemic amyloidosis and how by blocking both IL-17A and IL-17F affects arteriosclerosis. These results seem more focused on inflammation in liver, spleen, kidney and aortic tissues, rather than proving that skin inflammation leads to systemic conditions. Without clear evidence of this connection, the link to skin inflammation feels weak.
I suggest reconsidering how much emphasis is placed on the skin in the title, abstract, and introduction. The real focus of your study seems not to be on the skin but on the liver, spleen, kidneys and aorta. Shifting the focus to better match your actual findings would make the study stronger and more accurate.
While i thank you for your response, i still think the manuscript should be revised to better reflect the data you’ve presented.
Response: Thank you very much for taking the time to review this report. It became statistically clear that intractable skin diseases such as psoriasis vulgaris and atopic dermatitis increase cerebrovascular and cardiovascular sclerosis, resulting in a higher rate of cerebral and myocardial infarctions and, consequently, a shorter life span. We were searching for the reason why arteriosclerosis is cheap to develop. We are also conducting human studies and research using mouse models to search for links.
Data also prove that psoriasis and atopic dermatitis are associated with a higher frequency of comorbidities and complications such as diabetes, osteoporosis, systemic amyloidosis, and liver fibrosis, which are also secondary to dermatitis.
By suppressing cutaneous inflammation, skin-derived cytokines and these sequelae can be prevented. There is evidence that the incidence of cerebrovascular disease and myocardial infarction can be reduced by anti-cytokine therapies such as anti-TNF-α inhibitors. We are currently conducting educational activities to promote the desirability of aggressive treatment of psoriasis and atopic dermatitis in order to prevent complications and comorbidities. However, there are still many unknowns regarding the pathogenesis of complications and comorbidities and which cytokines play a key role, and we need to accumulate more data.
In the past, KCASP1Tg mice were used to show that IL-18 is released from the skin upon skin breakdown, resulting in massive production of nonspecific IgE, which later led to the concept of innate lymphoid cell type 2. He also found that the crossbreeding of these mice with IL-18KO inhibited IgE production, and that the extremely increased IgE in atopic dermatitis was caused by the release of IL-18 due to the high level of nonspecific IgE and skin breakdown (Nature immunology. 2000; 1 (2) : 132-7; 1 (2) : 132-7). 1 (2): 132-7. ). In addition, as described in the introduction of this article, we have reported that various complications, such as organ damage, occur after the onset of skin symptoms in KCASP1Tg mice. In other words, dermatitis triggers all systemic inflammation (Journal of Allergy and Clinical Immunology. Volume 136 Issue 3 Pages 823-824 ). This phenomenon is not limited to the KCASP1Tg mouse, but also to the KIL-18Tg mouse model of atopic dermatitis, which has milder inflammation than the KCASP1Tg mouse and expresses IL-18 specifically in the skin only, but also develops the same internal amyloidosis and atherosclerosis ( PLoS One. 2014 Aug 13;9(8):e104479. doi: 10.1371/journal.pone. 0104479.). Thus, the association between skin inflammation and visceral disease is clearer than previously reported, and this report is a continuation of these findings. Noteworthy findings of this paper are that inhibition of IL-17A and IL-17F, which markedly improve skin symptoms in the field of psoriasis, suppressed atherosclerosis, and that inhibition of IL-17A is essential for improvement of visceral amyloidosis, while inhibition of IL-17F is not required for improvement of visceral amyloidosis. This is a finding that is likely to attract a great deal of attention. We have changed the content of the text to make it more understandable. Thank you for your attention.
Comment: Also, about the ARRIVE guidelines the authors must add some more things in their materials and methods part. Especially
- Sample size justification (Why 5 animals and not less or more?)
- Weight of the animals (Providing also SD)
- Randomization methods (If not state that here was no randomization)
- Monitoring of animal’s health (for example Weight or temperature measurements throughout the experiment)
Response: We have made the changes you indicated.
Round 3
Reviewer 3 Report
Comments and Suggestions for Authors
Dear Authors,
I have reviewed your manuscript again, but I still find that your introduction does not clearly explain the rationale behind your experimental design as you outlined in our correspondence. Specifically, if you are certain that in this experimental model dermatitis precedes and subsequently leads to pathology in internal organs, then this should be explicitly stated in the introduction. In my view, the current manuscript does not clearly convey your perspective on this matter.
Additionally, information regarding the animal model you used, such as the timing of skin inflammation onset, should be in a separate paragraph from the experimental design. This way, readers can easily distinguish between the model characteristics and the design elements.
Regarding the ARRIVE guidelines, there are still mandatory elements missing:
- The weight of the animals and its variability.
- The biostatistical method used to determine the sample size. A general mention of ethical considerations is insufficient because the ethics of an experiment depend on the sample size choice. For example, using only two animals would not be ethically sound as it would likely yield inconclusive results, indicating that such a choice is unsupported.
- Lastly, you mentioned having 10 animals but only using 5. If this selection was not random (as you stated), then please specify the criteria used for this choice.
I suggest that the readability needs improvement.
Author Response
Comment: Dear Authors,
I have reviewed your manuscript again, but I still find that your introduction does not clearly explain the rationale behind your experimental design as you outlined in our correspondence. Specifically, if you are certain that in this experimental model dermatitis precedes and subsequently leads to pathology in internal organs, then this should be explicitly stated in the introduction. In my view, the current manuscript does not clearly convey your perspective on this matter.
Response: We have made the changes as the reviewer indicated.
Additionally, information regarding the animal model you used, such as the timing of skin inflammation onset, should be in a separate paragraph from the experimental design. This way, readers can easily distinguish between the model characteristics and the design elements.
Response: In the material and methods section, we have described the animal model and the experimental design separately.
Regarding the ARRIVE guidelines, there are still mandatory elements missing:
- The weight of the animals and its variability.
Response: We have made the changes as the reviewer indicated.
- The biostatistical method used to determine the sample size. A general mention of ethical considerations is insufficient because the ethics of an experiment depend on the sample size choice. For example, using only two animals would not be ethically sound as it would likely yield inconclusive results, indicating that such a choice is unsupported.
Response: We have made the changes as the reviewer indicated.
- Lastly, you mentioned having 10 animals but only using 5. If this selection was not random (as you stated), then please specify the criteria used for this choice.
Response: We did not randomly divide each group into several groups, but each group kept 10 animals, of which 5 were randomly used in the experiment. We have made the changes.
Round 4
Reviewer 3 Report
Comments and Suggestions for Authors
Dear authors
I am pleased to recommend your work for pubblication, after carrying out the amendements sugested.
Best regards
Comments on the Quality of English LanguageModerate language improvements may be considered